# Sex-based differences in front crawl and butterfly sprint performance in age-group swimmers

Radomyos Matjiur[1], Phornpot Chainok[1]*, Jessy Lauer[2,3], Weerawat Limroongreungrat[4], Karla de Jesus[5], Rodrigo Zacca[6,7,8,9], Ricardo J. Fernandes[10], J. Paulo Vilas-Boas[10]

**1** Faculty of Sport Science, Burapha University, Chonburi, Thailand, **2** Brain Mind Institute, School of Life Sciences, Swiss Federal Institute of Technology (EPFL), Lausanne, Switzerland, **3** Rowland Institute at Harvard, Harvard University, Cambridge, Massachusetts, United States of America, **4** College of Sport Science and Technology, Mahidol University, Thailand, **5** Human Performance Laboratory, Faculty of Physical Education and Physiotherapy, Federal University of Amazonas, Brazil, **6** Research Center in Physical Activity, Health and Leisure (CIAFEL), Faculty of Sports, University of Porto, Porto, Portugal, **7** Laboratory for Integrative and Translational Research in Population Health (ITR), Porto, Portugal, **8** Nucleus of Research in Human Motricity Sciences, Universidad Adventista de Chile, Chillán, Chile, **9** Laboratory of Sport Physiology, Faculty of Sports, University of Porto, Porto, Portugal, **10** Centre of Research, Education, Innovation and Intervention in Sport (CIFI$_2$D) and Porto Biomechanics Laboratory (LABIOMEP-UP), Faculty of Sport, University of Porto, Porto, Portugal

☙ These authors contributed equally to this work.
* phornpot@go.buu.ac.th

## Abstract

### Objectives

To identify the key variables influencing 50 m sprint performance in front crawl and butterfly strokes and determine the predominant predictors of sex-related performance differences in each technique in age-group swimmers.

### Methods

Thirty national-level age-group swimmers (15 boys and 15 girls; age 13.5 ± 1.0 years) underwent assessments of anthropometry, biological maturation, neuromuscular performance (dynamic strength index of upper and lower limbs), anaerobic critical velocity, tethered swimming force, and stroke mechanics.

### Results

Multivariate analyses revealed substantial sex differences ($p < 0.05$), with boys outperforming girls in upper-body force output and anaerobic critical velocity (AnCV). Boys showed a greater AnCV (1.54 vs. 1.45 m·s⁻¹) and lower fatigue index (−8.52% in front crawl). Boys and girls had a 27.60 N front crawl force differential and 13.59 N butterfly force difference in the first 10 s of tethered swimming. Controlling for sex and biological maturity, the maximum explained variation ($\eta^2$) was found in front crawl tethered swimming force (47%), and butterfly lower limb neuromuscular performance

**Data availability statement:** All relevant data are within the paper and its Supporting Information files.

**Funding:** Faculty of Sport Science, Burapha University THAILAND. The funders had no role in study design, data collection and analysis, decision to publish, or preparation of the manuscript.

**Competing interests:** There are no competing interests to report.

(29%). In front crawl, total push-up peak force (diff: 215.44 N; 95% CI: 1.25 to 3.08; $p < 0.001$) and body height (diff: 0.08 cm; 95% CI: −1.82 to −0.28; $p < 0.001$) were key discriminators, while in butterfly, push-up force (diff: 215.44 N; 95% CI: 1.25 to 3.08; $p < 0.001$) and stroke index (diff: 0.54 m²·s⁻¹; 95% CI: 1.04 to 2.79; $p < 0.001$) were the most effective predictors.

## Conclusion

Our findings confirm the predictive ability of upper-limb neuromuscular performance and stroke efficiency in identifying sex-based differences in sprint swimming. The practical implications of these findings are substantial, offering a framework for coaches and sports scientists to improve procedures for training by focusing on technique- and sex-specific performance characteristics at critical stages of athletic development.

## Introduction

Swimming is predominantly considered as a biophysical sport, where biomechanical and bioenergetic factors play pivotal roles in determining performance. Consequently, coaches and swimming scientists often prescribe training and evaluate physical conditioning and technical proficiency in combination, instead of assuming independent analysis [1,2]. The efficacy of swimming, especially in sprint events such as the 50 m simultaneous (butterfly), and alternated technique (front crawl), is significantly determined by a swimmer's ability to generate force and maintain velocity throughout short distances [3–5]. While factors including swimming technique, starts and turns [6,7], along with physical and physiological endurance and energy system capacity and power are essential [1,2], the coordinated generation of propulsive force, involving strength and power in both the upper and lower limbs, is crucial in determining performance success.

Previous investigations into the variables and characteristics of propulsive force associated with swimming performance have predominantly focused on indirect assessments of land-based strength and conditioning, encompassing isokinetic evaluations, one-repetition maximum (1-RM), countermovement jump (CMJ), isometric peak torque and handgrip strength [8–14]. Direct assessments of a swimmer's maximal strength and force production capabilities have previously involved fully-tethered [11,15] and semi-tethered swimming [16,17], alongside evaluations of anaerobic and sprint performance across four swimming techniques [9,18,19]. Still, although tethered swimming provides a direct assessment of force production [11], current research has not established the optimal balance between maximal strength and explosive power that facilitates faster and more efficient techniques, thus enhancing sprint swimming performance.

Sprint swimming is an event where variations in movement techniques, both simultaneous (breaststroke and butterfly) and alternated (front crawl and backstroke), may be emphasized and influenced by distinct primary biophysical factors, including

energy and biomechanical aspects [20]. From a biomechanical perspective, the ability to enhance mechanical and propulsive efficiency could be associated with the optimization of swimming technique and the capacity to balance maximal and explosive strength, referred to as the dynamic strength index (DSI) [3,9,21,22]. Considering that DSI profiling is probably associated with the 50 m sprint swimming distance, the correlations between swimming performance and the profile components of the upper and lower limbs may differ separately between simultaneous and alternating techniques.

Consequently, biophysical factors, encompassing energy and biomechanical aspects, should not be independently examined. The complex relationship between upper and lower limb neuromuscular performance, force generation during fully-tethered swimming, anaerobic critical velocity (AnCV) and swimming performance at 50 m remains controversial, with no consensus regarding the bilateral and unilateral limbs that directly influence the biomechanics of sprint swimming particularly in age-group. Notwithstanding the apparent inconsistencies in some available results, there is a compelling imperative to enhance the development of age-group swimmers, particularly during the pubertal period, when maturity status can significantly influence a swimmer's growth, development and performance [23]. Therefore, the purposes of this study were: (i) to conduct a multivariate analysis of age-group swimmers' performance to determine which variables are predominant to achieve better performances when analyzing the difference between sex in two swimming techniques (front crawl and butterfly) and (ii) identify the least set of predictors that discriminate the sex differences.

With this in mind, we also intend to provide important and useful information that may help coaches, strength and conditioning and sport scientists to improve the development and understanding characteristics of propulsive force associated with swimming performance, and increase success opportunities in their training sessions and competitions. Besides, we hypothesized that differences in swimming performance across techniques are associated with a minimal set of key physiological and biomechanical predictors, specifically AnCV, tethering force, and upper and lower limb dynamic strength index (DSI).

## Materials and methods

### Participants

A priori power analysis was conducted using G*Power version 3.1.9.2 to determine the required sample size. The analysis indicated that a minimum of 30 participants was necessary to detect a large effect size (0.95) with a statistical power of 0.95 and an alpha level of 0.05. Accordingly, thirty national-level age-group swimmers (15 boys and 15 girls), all specializing in short-distance events and having provided written informed consent, were recruited for the study. All participants, possessed at least of three year of competition experience in regional or national competitions, underwent current medical evaluations and qualified for the national championships. Swimmers with any contraindications for study participation were excluded. No swimmers in the age group were utilizing medications or dietary supplements recognized to affect physical performance. The research complied with the standards set out by the World Medical Association's Declaration of Helsinki, and ethics approval was granted by the local university (code number HS025/2567(C3)). The participants and their guardians were provided with a comprehensive explanation of the testing protocols before obtaining signed consent for participation.

### Testing procedures

We conducted a prospective cohort study using descriptive, comparative, correlational, and multivariate analysis to assess physiological variables (i.e., AnCV), biomechanical factors (e.g., tethering force and stroke mechanics — stroke length [SL], stroke rate [SR] and stroke index [SI]) and neuromuscular performance (i.e., DSI of upper and lower limbs) affecting performance across sex and techniques. Testing was conducted during the final four weeks leading up to the National Age–Group Championships. Assessments were carried out on five separate occasions between the first macrocycle (January -April, 2025) of a traditional three-peak preparation program.

On the first visit, participants underwent a comprehensive assessment of their anthropometric data, body composition, and biological maturation. Subsequently, on the same day, a second testing session was executed, comprising a maximal 50 m front crawl swimming assessment in a 25 m pool. On the following day, the dynamic strength index (DSI) for both the upper and lower limbs was assessed. On the third day, the anaerobic critical velocity (AnCV) was assessed with maximum effort trials of 10 m, 15 m and 25 m in front crawl and butterfly, with a 30 min rest period between techniques. The fourth day included a 30 s maximal tethered swimming force assessment, conducted with a specialized tethered swimming force apparatus (Swimforce V1.0.0, Germany) for both front crawl and butterfly techniques, with a 30 min interval between the trials. Finally, an all-out 50 m butterfly swimming performance was performed in a 25 m pool at the final appointment.

### Anthropometry, body composition and biological maturation

The standing height was measured using ultrasonic height meter in which automatically measures height for improved input accuracy (X-Contact 357S, Jawon Medical Co., Ltd., Korea) and body mass and body mass index (BMI; $kg \cdot m^{-2}$) were evaluated using bioelectrical impedance analysis (BIA; Body Composition Analyzer: X-Contact 357S, Jawon Medical Co., Ltd., Korea), performed by a qualified anthropometrist with over fifteen years of experience, ensuring compliance with established anthropometric measurement protocols according to the techniques described by the International Society for the Advancement of Kinanthropometry [30]. The biological maturity offset was determined by calculating the age at peak height velocity (age-at-PHV) [24]. Weight, height, seating height and sex were all factors in a specific calculation. The maturity offset is a numerical value that measures how many years an individual is away from their peak height velocity (PHV) age. A maturity offset can be positive (+) or negative (-) showing how many years the participant has been in the sport after reaching PHV and how many years the participant has to go before reaching PHV, respectively.

### Neuromuscular performance

The lower-limb dynamic strength index (DSI) was calculated as the ratio of peak force during the countermovement jump (CMJ) to peak force during the isometric mid-thigh pull (IMTP), based on data extracted from their respective force-time curves [25]. The CMJ were evaluated at a 1000 Hz sample rate utilizing the commercially available K-deltas dual force platforms (Kinvent Physio, Montpellier, France). Peak force measurements from these evaluations encompassed dominant CMJ peak force (N), non-dominant CMJ peak force (N), and total CMJ peak force (N), which were utilized for further investigation. The isometric mid-thigh pull (IMTP) was performed on the K-deltas dual force platforms 60 s subsequent to the CMJ. In this test, age-group swimmers exerted maximal force by pulling a bar and pressing their feet against the force platform for 5 seconds. Peak force values, including dominant and non-dominant IMTP peak force (N), and total IMTP peak force (N) were derived from the force-time curves for subsequent analysis.

Peak forces (N) from both the CMJ and the IMTP were used for subsequent analysis of the lower limb dynamic strength index (DSI). To comprehensively assess strength characteristics, the unilateral strength asymmetry index was also calculated for both CMJ and IMTP, which serves as an evaluation of maximal isometric force. This index was determined by computing the percentage difference between the dominant (D) and non-dominant (ND) limbs. The dominant limb was methodically defined as the one that exhibited the higher peak force during the test. The asymmetry index was calculated using the following formula: $(D - ND)/ D \times 100$ [22].

For the component of upper limb DSI, ballistic push-up and isometric grip strength were used to determine upper extremity DSI from K-deltas dual force platforms (Kinvent Physio, Montpellier, France) [25]. The ballistic push-up test was performed at a sampling rate of 1000 Hz using the K-deltas dual force platforms (Kinvent Physio, Montpellier, France). Age-group swimmers positioned their hands at approximately shoulder width on a K-deltas dual force platform

and lowered their chests until contact with the plate was made [25]. The peak force measurements from this test, including Dominant push-up peak force (N), Non-dominant push-up peak force (N), and Total push-up peak force (N), were recorded for subsequent analysis. [25].

The isometric grip strength was measured using a K-Grip Dynamometer (Kinvent Physio, Montpellier, France) for a duration of 5 s, 60 s after completing the ballistic push-up, with a 30 s rest time between each side. The peak force values recorded were dominant grip strength peak force (N), non-dominant grip strength peak force (N) and total grip strength peak force (N). Peak forces (N) from the ballistic push-up and static grip strength tests were utilized for subsequent analysis for upper limb DSI. The unilateral strength asymmetry index of the ballistic push-up and static grip strength test was determined as the percentage difference between the dominant limb and the non-dominant limb as proposed by Bishop et al. [22].

## Tethered swimming force

The tethered swimming force was measured using a load-cell device (Swimforce V1.0.0, Germany), which recorded data at a frequency of 100 Hz and had a maximum capacity of 1,000 N. The system was situated on a starting block 5 m distance, sloping at an inclination of 5.7° to the water surface [26]. The maximal tethered swimming tests for the whole crawl and butterfly technique were conducted in separate testing sessions. Force data was consistently gathered during a 30 s maximal effort interval. The data was subsequently imported into Acqknowledge 4.0 (BIOPAC Systems, Inc., CA, USA) and filtered with a 15 Hz cut-off digital filter (finite impulse response, FIR – Window Blackman −61 dB), as established through Fast Fourier Transform (FFT) analysis to reduce artifact noise [26]. The key force variables derived from the individual force-time curves included the mean force at each 10 s interval and the fatigue index [26,27].

## Anaerobic critical velocity (AnCV)

Anaerobic performance for front crawl and butterfly swimming techniques was assessed using the AnCV method that originally adapted for swimming from the critical power concept that provides a reliable measure of an athlete's anaerobic capacity [28,29]. This approach, widely recognized in swimming research which was calculated for each age-group swimmer by analyzing the slope of the distance-time (Dd-t) relationship, based on swimming performance times for 10, 15 and 25 m over time [28,29]. All swimming performance times for the 10 m, 15 m, and 25 m trials were meticulously recorded by two experienced researchers using Seiko S140 chronometers. To ensure exceptional inter-observer reliability, the final time utilized was the average of the two recorded measures, based upon their values not differing by more than 0.20 seconds [28,29]. A 10-minute rest period was strictly enforced between each swimming session to ensure sufficient recovery. The regression model is expressed as $y = ax + b$, where y denotes the distance covered and x represents time. In this context, the coefficient an is primarily recognized as the short-distance velocity coefficient (slope), while also being recognized for its conventional meaning in the literature as part of the anaerobic critical velocity (AnCV). The constant b corresponds to the y-intercept of the regression line [28,29].

## Swimming performance

Age-group swimmers completed a 400 m warm-up swim, followed by a 15 min passive rest period prior to the 50 m front crawl and butterfly block all-out performance time trial. The 50 m race times were recorded by qualified timekeeper using a SEIKO S056 stopwatch (Tokyo, Japan). Results were obtained for three key kinematic variables frequently discussed in swimming biomechanics: stroke rate (SR), stroke length (SL), and stroke index (SI). SR and SL were calculated from the time it took to complete three consecutive stroke cycles. Stroke length was determined using the ratio of swimming velocity (v) to the corresponding SR, while SI was calculated by multiplying the swimming velocity (v) by the SL [30].

## Statistical analysis

Descriptive statistics are presented as mean ± standard deviation (M ± sd). The normality assumption was assessed using the Shapiro–Wilk test, where no significant violations were noticed. Sex differences were firstly compared in a univariate fashion using Student $t$ test; Cohen's $d$ values were also calculated and interpreted as follows: < 0.20 (trivial), 0.20 to 0.59 (small), 0.60 to 1.19 (moderate), 1.20 to 1.99 (large), 2.0 to 3.9 (very large), and > 4.0 (extremely large) [31]. Then, the effects of sex and biological maturation on anthropometric and body composition, neuromuscular performance (upper and lower limbs DSI), tethering force, AnCV and swimming technique kinematics (SI) were examined using a multivariate analysis of covariance (MANCOVA) and eta squared ($\eta^2$) was used as a measure of explained variance. Further, using only variables that had a statistically significant effect, a forward stepwise discriminant function analysis was employed to identify the smallest set of variables that maximizes the differences between sexes for each swimming technique. The statistical analyses were conducted using SPSS 25.0 (IBM Corp., Armonk, New York, US), significance level was established at alpha = 0.05.

## Results

Descriptive statistics for age, anthropometric and body composition, biological maturation, neuromuscular performance, AnCV, tethered force and swimming performance variables of boys and girls age-group swimming in front crawl and butterfly swimming technique are shown in Table 1 and Table 2, respectively. Substantial sex differences were identified in the selected variables ($p < 0.05$), with the exception of the CMJ asymmetry index, ITMP asymmetry index, lower limb DSI, push-up asymmetry index, grip strength asymmetry index and upper limb DSI in both swimming techniques, as well as AnCV and SR for butterfly.

When controlling for biological maturation, the MANCOVA results (Table 3) revealed significant results for each variable set only in the swimming tethered force. A total of 21 variables indicated that boys swimmers outperformed girls swimmers, with $\eta^2$ ranging from 23% (anthropometry and body composition) to 47% (tethered swimming force). For butterfly, the MANCOVA results (Table 4) revealed no significant results across all multivariate tests with a total of 20 variables in which boys outperformed girls swimmers and the $\eta^2$ varied from 7% (tethered swimming force) to 29% (lower limbs neuromuscular performance).

Table 5 reports the main results of the forward stepwise discriminant function, and shows the best smaller set of the previous twenty-one variables that best discriminates the sex and are in order of importance of total push-up peak force and body height in front crawl and total push-up peak force and stroke index from the best smaller set of the previous twenty variables in butterfly, respectively.

## Discussion

This study investigated the multivariate profiles of age-group swimmers by sex in front crawl and butterfly techniques, controlling for biological development, and identified the minimal set of predictors that differentiate sexes. This study provided comprehensive evidence that growth and maturation, AnCV, tethering force and the DSI of both upper and lower limbs play pivotal roles in determining 50 m sprint swimming performance in age-group swimmers. Multivariate analysis revealed significant sex-based differences across physiological and biomechanical variables, with boys swimmers outperforming girls in several key domains, even after controlling for biological maturity. Total push-up peak force and body height distinguished sex differences in front crawl, while total push-up peak force and stroke index differentiated sex in butterfly. Thus, these findings offer support to the proposed hypotheses and also highlight the complexity of performance determinants across swimming techniques.

The present study, considering sex differences, demonstrated that boys outperformed girls on multiple anthropometric, neuromuscular, and tethered-force characteristics regularly observed in research on young swimmers [2,3,32]. In along with describing these expected differences, our findings enhance existing knowledge by identifying the most significant

**Table 1. Descriptive statistics (M ± SD) for general characteristics, anthropometry and maturation and neuromuscular performance in age-group swimmers.**

| Variables | Boys | Girls | Mean difference (95% CI) | Cohen's d |
|---|---|---|---|---|
| | M ± SD | M ± SD | | |
| Age (years) | 13.27 ± 0.33 | 13.73 ± 0.96 | −0.47 (−1.23, 0.23) | −0.51 |
| 50 m World aquatic point | 418.80 ± 86.12 | 454.60 ± 87.83 | −35.38 (−1.13, 0.32) | −0.41 |
| Biological maturation | | | | |
| Maturity offset (years) | −1.37 ± 0.88 | −0.43 ± 0.97 | −0.93(−1.82, −0.28)* | −1.01 |
| Anthropometry and body composition | | | | |
| Height (cm) | 1.70 ± 0.07 | 1.62 ± 0.05 | 0.08 (−1.82, −0.28)** | 1.44 |
| Body mass (kg) | 60.73 ± 9.23 | 50.00 ± 4.33 | 10.73 (0.62, 2.34)** | 1.48 |
| Body mass index (kg·m$^{-2}$) | 20.82 ± 2.37 | 19.08 ± 1.60 | 1.74 (0.10, 1.60)* | 0.86 |
| Lower limbs neuromuscular performance | | | | |
| Dominant CMJ peak force (N) | 750.83 ± 119.84 | 552.47 ± 63.24 | 198.37 (1.16, 2.96)** | 2.07 |
| Non-dominant CMJ peak force (N) | 704.65 ± 118.12 | 524.52 ± 56.66 | 180.12(1.06, 2.81)** | 1.94 |
| Total CMJ peak force (N) | 1,455.52 ± 236.26 | 1,076.99 ± 116.95 | 378.53(1.13, 2.91)** | 2.03 |
| CMJ asymmetry index (%) | 6.22 ± 3.84 | 4.91 ± 4.55 | 1.31 (−0.41, 1.03) | 0.31 |
| Dominant ITMP peak force (N) | 958.47 ± 273.36 | 712.21 ± 132.76 | 246.27(0.36, 1.91)* | 1.15 |
| Non-dominant ITMP peak force (N) | 876.71 ± 242.81 | 633.31 ± 119.65 | 243.40(0.47, 2.05)** | 1.27 |
| Total IMTP peak force (N) | 1,835.17 ± 511.89 | 1,344.86 ± 251.28 | 490.31(0.42, 1.99)** | 1.22 |
| ITMP asymmetry index (%) | 8.39 ± 5.88 | 11.13 ± 3.70 | −2.73 (−1.28, 0.18) | −0.56 |
| Lower limb DSI (%) | 0.85 ± 0.27 | 0.82 ± 0.12 | 0.04 (−0.55, 0.89) | 0.17 |
| Upper limbs neuromuscular performance | | | | |
| Dominant push-up peak force (N) | 369.01 ± 67.83 | 257.10 ± 37.93 | 111.91 (1.13, 2.92)** | 2.04 |
| Non-dominant push-up peak force (N) | 328.06 ± 57.68 | 224.52 ± 36.17 | 103.54 (1.23, 3.05)** | 2.15 |
| Total push-up peak force (N) | 697.07 ± 122.42 | 481.63 ± 68.05 | 215.44 (1.25, 3.08)** | 2.18 |
| Push-up asymmetry index (%) | 11.61 ± 7.80 | 12.11 ± 10.25 | −0.49 (−0.77, 0.66) | −0.05 |
| Dominant grip strength peak force (N) | 304.06 ± 66.07 | 220.67 ± 44.17 | 83.39 (0.66, 2.29)** | 1.48 |
| Non-dominant grip strength peak force (N) ()(N)(N) | 282.29 ± 65.45 | 201.62 ± 46.40 | 80.67 (0.61, 2.21)** | 1.42 |
| Total grip strength peak force (N) | 586.35 ± 130.37 | 422.29 ± 90.06 | 164.07 (0.64, 2.27)** | 1.46 |
| Grip strength asymmetry index (%) | 7.43 ± 6.01 | 9.03 ± 5.91 | −1.67 (−0.99, 0.44) | −0.28 |
| Upper limb DSI (%) | 1.24 ± 0.31 | 1.17 ± 0.21 | 0.07 (−0.47, 0.97) | 0.25 |

*Note*: CMJ = Counter movement jump; IMTP = Isometric mid-thigh pull; AnCV = Anaerobic critical velocity DSI = Dynamic strength index; (NS) = Non-significant; (*) = p < 0.05; (**) = p < 0.01; (***) = p < 0.001.

elements that differentiate sex-specific sprint characteristics. Beyond confirming expected sex differences, our findings enhance current knowledge by identifying the principal elements that distinctly differentiate sex-specific sprint features. Utilizing a multivariate model that incorporates growth and maturation [24], upper- and lower-limb DSI [21,22], short-distance velocity coefficients from the AnCV [28,29], tethered-force outputs [11,15] and stroke kinematics [23], we determined a minimal but highly discriminative set of variables that effectively differentiates sprint performance characteristics between boys and girls.

From the forward stepwise discriminant analysis it was identified that total push-up peak force was the strongest discriminant variable in both front crawl and butterfly techniques, highlighting the critical importance of upper-body explosive strength in short-distance sprint swimming. This metric likely reflects a combination of muscular power, interlimb coordination and neuromuscular control, all of which are essential for effective propulsion [3,9,13]. Moreover, in front crawl,

**Table 2. Descriptive statistics (M±SD) for anaerobic critical velocity (AnCV), tethered force and swimming performance in age-group front crawl and butterfly swimmers.**

| Variables | Boys | Girls | Mean difference (95 % CI) | Cohen's d |
|---|---|---|---|---|
| | M ± SD | M ± SD | | |
| **Front crawl** | | | | |
| Anaerobic Critical velocity | | | | |
| AnCV (m·s⁻¹) | 1.54 ± 0.09 | 1.45 ± 0.07 | 0.08 (0.26, 1.80)** | 1.04 |
| Swimming tethered force | | | | |
| Average mean force (0-10s) (N) | 78.92 ± 19.10 | 51.33 ± 15.70 | 27.60 (0.74, 2.39)** | 1.58 |
| Average mean force (10-20s) (N) | 65.77 ± 16.37 | 42.49 ± 11.75 | 23.29 (0.79, 2.46)** | 1.63 |
| Average mean force (20-30s) (N) | 53.33± 14.40 | 31.73 ± 9.27 | 21.59 (0.92, 2.63)** | 1.78 |
| Fatigue index (%) | 36.62± 10.21 | 45.13 ± 8.31 | -8.52 (-1.66, -0.15)* | -0.92 |
| Swimming performance | | | | |
| Average velocity (m·s⁻¹) | 1.78 ± 0.13 | 1.65 ± 0.07 | 0.13 (0.44, 2.02)** | 1.24 |
| Stroke rate: SR (cycles·min⁻¹) | 50.33 ± 3.22 | 47.84 ± 2.48 | 02.49 (0.11, 1.66)* | 0.87 |
| Stroke length: SL (m) | 2.12 ± 0.17 | 2.07 ± 0.14 | 0.05 (-0.39, 1.06)** | 0.34 |
| Stroke index: SI (m²·s⁻¹) | 3.78 ± 0.52 | 3.41 ± 0.33 | 0.38 (0.10, 1.60)* | 0.86 |
| **Butterfly** | | | | |
| Anaerobic Critical velocity AnCV (m·s⁻¹) | 1.49 ± 0.14 | 1.42 ± 0.09 | 0.08 (-0.09, 1.38) | 0.65 |
| Anaerobic Critical velocity AnCV (m·s⁻¹) | 1.49 ± 0.14 | 1.42 ± 0.09 | 0.08 (-0.09, 1.38) | 0.65 |
| Swimming tethered force | | | | |
| Average mean force (0-10s) (N) | 64.15 ± 15.80 | 50.56 ± 12.73 | 13.59 (0.18, 1.70)* | 0.95 |
| Average mean force (10-20s) (N) | 53.81 ± 12.21 | 39.09 ± 5.93 | 14.72 (0.70, 2.34)** | 1.53 |
| Average mean force (20-30s) (N) | 46.64 ± 11.56 | 32.34 ± 3.93 | 14.29 (0.81, 2.48)** | 1.66 |
| Fatigue index (%) | 32.61 ± 16.18 | 39.37 ± 8.13 | -6.75 (-1.25, 0.21)** | -0.53 |
| Swimming performance | | | | |
| Average velocity (m·s⁻¹) | 1.62 ± 0.16 | 1.51 ± 0.07 | 0.10 (0.07, 1.57)* | 0.83 |
| Stroke rate: SR (cycles·min⁻¹) | 53.27 ± 6.57 | 54.36 ± 4.89 | -1.07 (-0.90, 0.53) | -0.19 |
| Stroke length: SL (m) | 1.83 ± 0.16 | 1.68 ± 0.12 | 0.15 (0.31, 1.85)** | 1.09 |
| Stroke index: SI (m²·s⁻¹) | 3.08 ± 0.36 | 2.53 ± 0.17 | 0.54 (1.04, 2.79)** | 1.93 |

*Note*: CMJ = Counter movement jump; IMTP = Isometric mid-thigh pull; AnCV = Anaerobic critical velocity DSI = Dynamic strength index; * = p < 0.05; ** = p < 0.01; *** = p < 0.001.

body height was a secondary predictor, suggesting the added mechanical advantage of longer limbs and SL in alternating techniques [32,34].

In contrast, SI was a key differentiator in butterfly, indicating the role of stroke efficiency and fluid coordination in simultaneous techniques, which require more precise motor timing and technical control [33,38]. Moreover, the stronger discriminant power of SI in butterfly indicates that stroke efficiency and motor coordination may have heightened importance in simultaneous techniques compared to alternating ones. Therefore, training strategies for young swimmers should not only emphasize strength development but also target technique refinement, particularly in butterfly, where technical execution appears more sensitive to differentiating performance [33].

Sprint swimming, encompassed by both simultaneous (breaststroke and butterfly) and alternating techniques (front crawl and backstroke), is determined by different important biophysical components, including energy and biomechanical elements [20]. Previous studies in young swimmers attempt to comprehend how anthropometric factors, growth and maturation and upper/lower limbs variables associated with swimming technique (biomechanics, energetics and efficiency)

**Table 3. Multivariate analyses of covariance (MANCOVA) with sex and maturity offset as covariates in the front crawl technique.**

| Variables | Multivariate Test | | Univariate Test | |
|---|---|---|---|---|
| | F | η² | F | η² |
| Anthropometry and body composition | 2.38 | 0.23 | | |
| Height (cm) | | | 23.12*** | 0.47 |
| Body mass (kg) | | | 21.38*** | 0.45 |
| Body mass index (kg·m⁻²) | | | 6.27* | 0.19 |
| Lower limbs neuromuscular performance | 0.82 | 0.29 | | |
| Dominant countermovement jump peak force (N) | | | 33.59*** | 0.56 |
| Non-dominant countermovement jump peak force (N) | | | 28.84*** | 0.53 |
| Total countermovement jump peak force (N) | | | 31.77*** | 0.55 |
| Countermovement jump asymmetry index (%) | | | 0.35 | 0.01 |
| Dominant isometric mid-thigh pull peak force (N) | | | 14.44** | 0.36 |
| Non-dominant isometric mid-thigh pull peak force (N) | | | 17.93*** | 0.41 |
| Total isometric mid-thigh pull peak force (N) | | | 16.33*** | 0.39 |
| Isometric mid-thigh pull asymmetry index (%) | | | 1.97 | 0.07 |
| Lower limb dynamic strength index (%) | | | 0.13 | 0.01 |
| Upper limbs neuromuscular performance | 0.88 | 0.27 | | |
| Dominant push-up peak force (N) | | | 49.88*** | 0.66 |
| Non-dominant push-up peak force (N) | | | 53.71*** | 0.77 |
| Total push-up peak force (N) | | | 57.9*** | 0.69 |
| Push-up asymmetry index (%) | | | 0.05 | 0.01 |
| Dominant grip strength peak force (N) | | | 23.19*** | 0.47 |
| Non-dominant grip strength peak force (N) | | | 21.73*** | 0.46 |
| Total grip strength peak force (N) | | | 22.88** | 0.47 |
| Grip strength asymmetry index (%) | | | 0.65 | 0.02 |
| Upper limb dynamic strength index (%) | | | 0.29 | 0.01 |
| Anaerobic Critical velocity (m·s⁻¹) | | | 14.94*** | 0.37 |
| Swimming tethered force | 4.37** | 0.43 | | |
| Average mean force (0-10s) (N) | | | 25.72*** | 0.50 |
| Average mean force (10-20s) (N) | | | 27.68*** | 0.52 |
| Average mean force (20-30s) (N) | | | 36.79*** | 0.59 |
| Fatigue index (%) | | | 15.69*** | 0.38 |
| Swimming Performance | 0.74 | 0.26 | | |
| Average velocity (m·s⁻¹) | | | 20.72*** | 0.44 |
| Stroke rate (cycles·min⁻¹) | | | 2.10 | 0.08 |
| Stroke length (m) | | | 7.90** | 0.23 |
| Stroke index: SI (m²·s⁻¹) | | | 18.38** | 0.44 |

*Note*: CMJ = Counter movement jump; IMTP = Isometric mid-thigh pull; AnCV = Anaerobic critical velocity DSI = Dynamic strength index; (NS) = Non-significant; (*) = p < 0.05; (**) = p < 0.01; (***) = p < 0.001.

influence performance [10,13,20,32]. The impact of development and maturation on athletic performance is well-documented, especially throughout adolescence, characterized by fast biological changes [35].

The present study reveals notable sex variations in anthropometric measurements, neuromuscular performance, AnCV and tethering force variables, even when adjusting for maturity, highlighting the developmental disparities between young boys and girls swimmers. Boys were taller and heavier than girls, with higher values of the body mass index, while being

**Table 4. Multivariate analyses of covariance (MANCOVA) with sex and maturity offset as covariates in the butterfly technique.**

| Variables | Multivariate Test | | Univariate Test | |
|---|---|---|---|---|
| | F | η² | F | η² |
| Anthropometry and body composition | 2.28 | 0.23 | | |
| Height (cm) | | | 23.12*** | 0.47 |
| Body mass (kg) | | | 21.38*** | 0.41 |
| Body mass index (kg·m⁻²) | | | 6.23* | 0.19 |
| Lower limbs neuromuscular performance | 0.82 | 0.29 | | |
| Dominant countermovement jump peak force (N) | | | 33.59*** | 0.56 |
| Non-dominant countermovement jump peak force (N) | | | 28.84*** | 0.53 |
| Total countermovement jump peak force (N) | | | 31.77*** | 0.55 |
| Countermovement jump asymmetry index (%) | | | 0.35 | 0.01 |
| Dominant isometric mid-thigh pull peak force (N) | | | 14.41** | 0.36 |
| Non-dominant isometric mid-thigh pull peak force (N) | | | 17.93*** | 0.41 |
| Total isometric mid-thigh pull peak force (N) | | | 16.33*** | 0.39 |
| Isometric mid-thigh pull asymmetry index (%) | | | 1.97 | 0.07 |
| Lower limb dynamic strength index (%) | | | 0.13 | 0.01 |
| Upper limbs neuromuscular performance | 0.88 | 0.27 | | |
| Dominant push-up peak force (N) | | | 49.88*** | 0.66 |
| Non-dominant push-up peak force (N) | | | 53.71*** | 0.67 |
| Total push-up peak force (N) | | | 57.96*** | 0.69 |
| Push-up asymmetry index (%) | | | 0.05 | 0.01 |
| Dominant grip strength peak force (N) | | | 23.19*** | 0.47 |
| Non-dominant grip strength peak force (N) | | | 21.73*** | 0.46 |
| Total grip strength peak force (N) | | | 22.88*** | 0.47 |
| Grip strength asymmetry index (%) | | | 0.65 | 0.02 |
| Upper limb dynamic strength index (%) | | | 0.29 | 0.01 |
| Anaerobic Critical velocity (m·s⁻¹) | | | 6.37* | 0.20 |
| Swimming tethered force | 0.40 | 0.07 | | |
| Average mean force (0-10s) (N) | | | 9.99** | 0.23 |
| Average mean force (10-20s) (N) | | | 15.77*** | 0.38 |
| Average mean force (20-30s) (N) | | | 19.33*** | 0.43 |
| Fatigue index (%) | | | 0.66 | 0.23 |
| Swimming Performance | 0.91 | 0.09 | | |
| Average velocity (m·s⁻¹) | | | 7.91** | 0.23 |
| Stroke rate (cycles·min⁻¹) | | | 0.04 | 0.01 |
| Stroke length (m) | | | 7.17* | 0.22 |
| Stroke index: SI (m²·s⁻¹) | | | 25.74*** | 0.50 |

*Note*: CMJ = Counter movement jump; IMTP = Isometric mid-thigh pull; AnCV = Anaerobic critical velocity DSI = Dynamic strength index; * = p < 0.05; ** = p < 0.01; *** = p < 0.001.

less advanced in their biological maturation. Some anthropometric characteristics may be especially beneficial in swimming, as high limb length and low body surface area can improve propulsion and decrease drag, particularly in sprint events where hydrodynamic and propulsive efficiency over short distances is crucial [32,34].

From the perspective of neuromuscular performance in sprint swimming, particularly in the 50 m freestyle and butterfly, understanding the interplay between explosive strength, limb-specific force production, and stroke mechanics is essential

**Table 5. Summary of stepping in forward stepwise discriminant analysis in front crawl and butterfly techniques.**

| Step | Entered | Wilks's Lambda | Approx. F-Ratio | p-Value |
|---|---|---|---|---|
| Front crawl | | | | |
| 1 | Total push-up peak force (N) | 0.818 | 64.43 | p<0.001 |
| 2 | Body height (cm) | 0.457 | 36.85 | p<0.001 |
| Butterfly | | | | |
| 1 | Total push-up peak force (N) | 0.791 | 56.93 | p<0.001 |
| 2 | Stroke index (m²·s) | 0.721 | 44.45 | p<0.001 |

for optimizing performance and tailoring training interventions. Such events necessitate explosive strength, high swimming frequency and efficient transference of force over the entire swimming cycle [3,21]. From this perspective, the results showed that boys consistently exhibited greater peak force outputs in both upper (e.g., push-ups and grip strength) and lower limbs (e.g., countermovement jump, isometric mid-thigh pull), along with enhanced tethered swimming force and anaerobic capacity, particularly in front crawl. This finding is supported by various studies indicating that dry-land strength and power variables, particularly explosive upper limb, correlate with swimming performance [5,9,12,26].

Interestingly, although considerable performance differences, particularly as measured by upper and lower limb DSI, boys had higher absolute force outputs than girls. However, the DSI values exhibited no differences across sexes, indicating that the equilibrium between maximal and explosive strength persists between sexes, even when absolute force capacity diverges. Higher DSI values among boys in this study suggest a superior ability to convert maximal strength into functional explosive actions. This is especially important in sprints, where reaction time (in take-off) and stroke efficiency (SI) are critical. Girls, while often biologically more mature in early adolescence, may exhibit lower DSI due to higher fat mass proportions and less neuromuscular drive efficiency, factors that impact the ability to express power rapidly. This supports the notion that DSI is a maturity-independent indicator of neuromuscular function, useful for evaluating performance capacity in youth athletes [35]. Besides, this finding corroborates the principle that continued participation in appropriately structured strength and conditioning programs throughout adolescence is essential for optimizing neuromuscular development [32,36,37].

Despite differences in absolute performance values, and strength and power outputs between boys and girls, this study identified no differences in upper (7.1–12.2%) and lower limb (6.1–10.9%) asymmetry indices. This set of evidence might suggest that bilateral strength balance is consistent across sexes at this stage of development. From a practical perspective, the preservation of interlimb symmetry across sexes is encouraging, particularly in a sport like swimming, where symmetrical movement is fundamental. This was interesting as excessive asymmetry, typically defined as interlimb differences exceeding 10–15%, has been associated with increased injury risk and impaired movement mechanics [22,35]. Furthermore, previous research indicated that in-water asymmetry is more closely associated with alterations in swimming technique generated by the aquatic environment than with muscular imbalances [14]. Further, in-water asymmetry is predominantly affected by technical adaptations to hydrodynamic forces and respiratory patterns rather than by muscular imbalances on dry land [38].

Tethered swimming force offers an in-depth evaluation of in-water propulsive strength based on the force-time curve, including average force at each 10 s interval and the fatigue index [27]. In the front crawl, the disparity in mean force between boys and girls varied from 27.60 N during the initial 10 s to 21.59 N in the last 10 s period. In butterfly, boys exhibited higher performance compared to girls, with a mean force difference increasing from 13.59 N in the initial 10 s to 14.29 N in the final 10 s, consequently confirming their stronger propulsive capability even after adjusting for biological maturity. Moreover, boys demonstrated a smaller fatigue index compared to girls in both front crawl (−8.52%) and butterfly (−6.75%), indicating greater muscle endurance and fatigue resistance, essential for maintaining technique and speed

throughout the latter portion of sprint efforts. The current findings demonstrate that boys exhibit a greater ability to produce and maintain high force levels than girls across all tethered variables, supporting previous research that established a strong relationship between tethered force and sprint swimming velocity, which necessitates high levels of coordinated bilateral muscular activation in both simultaneous and alternating techniques [26,27].

From a physiological perspective in sprint swimming, AnCV serves as a crucial metric of anaerobic energy systems, especially significant in age-group athletes, where rapid developmental changes influence swimming performance. In the current study, sex difference in AnCV was observed during front crawl performance, with boys demonstrating higher values than girls (1.54 vs. 1.45 m·s⁻¹), while no such difference was evident in the butterfly technique (1.49 vs. 1.42 m·s⁻¹). This finding underscores the technique-specific nature of energetic demand and mechanical efficiency in sprint swimming [28,29,40].

The higher AnCV values in boys, despite their comparatively lower average biological maturity relative to girls in this age group, indicate a sex-specific advantage in the recruitment and usage of energy systems pertinent to the front crawl. This may result from a combination of neuromuscular efficiency, increased lean muscle mass and biomechanical efficacy in producing and maintaining propulsion [39,40]. Furthermore, the lack of notable sex disparities in AnCV for butterfly may indicate the higher technical and metabolic requirements of this technique, which can diminish the performance advantages associated only to muscle strength. This was an interesting result since butterfly swimming necessitates enhanced coordination, symmetrical force application, and precise timing, thereby diminishing the sex performance disparity throughout early developmental phases [33,39].

Notably, the current findings provide numerous practical implications for coaches and sports scientists engaged with age-group sprint swimmers. For boys, strength and conditioning for boys should focus on enhancing upper-body explosive strength, as peak push-up force was the most significant predictor of sprint performance in both strokes. Because boys exhibit significantly greater absolute force and propulsive capability, programs should integrate specialized power-focused workouts. These modalities directly improve rapid force generation and upper-limb propulsion, proving them very useful for optimizing short-distance sprint performance. Furthermore, the significant impact of body height on front crawl performance indicates that taller boys might derive benefits from training focused on extending stroke length and optimizing distance per stroke, consequently enhancing biomechanical efficiency in high-speed swimming.

The findings suggest that for girls, enhancing stroke efficiency especially the SI in butterfly could result in more significant performance improvements than focusing exclusively on force generation. Girl showed similar DSI values but lower absolute force outputs, indicating that technical skill may compensate for strength limitations in early adolescence. Consequently, coaches have to prioritize technique-driven sessions, incorporating propulsion timing drills, underwater video feedback, and rhythm-coordination progressions that enhance simultaneous arm-leg synchronization in butterfly stroke. Importantly, both sexes require a balanced program of dry-land and aquatic training, as upper-body neuromuscular strength and technical proficiency influence sprint performance. Integrating intensive upper-body strength training with SR and SL interaction exercises can enhance front crawl propulsion. Butterfly training should emphasize timing, streamlined control, and bilateral force symmetry. These instructions transform multivariate data into approaches for developing sex-responsive sprint swimming.

Despite this study provides valuable insights by analysing the multivariate profiles of age-group swimmers by sex in front crawl and butterfly techniques, controlling for biological development and identifying a minimal set of sex-differentiating predictors, it is crucial to recognize several limitations that may affect the interpretation of the findings and influence future research directions. First, sample size was rather small (n = 30), yet sufficient for the present multivariate statistical analyses. However, it is suggested to consider, in further studies, a larger sample size to improve the applicability of the findings to wider populations, encompassing various age demographics and competitive tiers. Second, although we controlled for biological maturation using maturity offset, this method provides only an estimation and may not perfectly reflect individual hormonal or skeletal maturity. The lack of direct assessment of pubertal status may introduce variance

in developmental status that affects performance metrics. Lastly, the use of dry-land neuromuscular assessments (e.g., push-up force, CMJ, IMTP) and their extrapolation to in-water performance must be interpreted cautiously since they do not fully capture the complex neuromuscular coordination and hydrodynamic demands of swimming.

## Conclusion

This study provides a comprehensive understanding of the multivariate factors influencing 50 m sprint swimming performance in age-group swimmers, emphasizing significant sex-specific physiological and biomechanical differences in front crawl and butterfly techniques. When adjusting for biological maturation, factors as AnCV, tethered swimming force and the DSI for both upper and lower limbs were found to affect sprint performance. Total push-up peak force consistently proved to be the most significant discriminant variable in both swimming techniques, highlighting the of upper-body explosive strength in short-distance competitions. Stroke-specific predictors, including body height in front crawl and SI in butterfly, underscore the distinct technical requirements inherent to each swimming technique. Boys, while exhibiting lower overall maturity levels, demonstrated superior propulsive strength, force maintenance and energetic capacity underscoring the significance of neuromuscular and energetic components in the development of sprint performance. These findings support training programs designed to encourage strength and power while prioritizing the development of stroke-specific techniques, especially in techniques necessitating high levels of motor coordination, especially the butterfly technique.

## Supporting information

**S1 Data. Descriptive data for general characteristics, swimming performance and tethered force in age-group front crawl and butterfly swimmers.**
(XLSX)

**S2 Data. Descriptive data for neuromuscular performance in age-group front crawl and butterfly swimmers.**
(XLSX)

## Author contributions

**Conceptualization:** Phornpot Chainok, Radomyos Matjiur, Rodrigo Zacca, Ricardo J. Fernandes, J. Paulo Vilas-Boas.

**Data curation:** Phornpot Chainok, Radomyos Matjiur.

**Formal analysis:** Phornpot Chainok, Radomyos Matjiur, Jessy Lauer, Karla de Jesus, Rodrigo Zacca.

**Funding acquisition:** Phornpot Chainok, Radomyos Matjiur.

**Investigation:** Phornpot Chainok, Radomyos Matjiur.

**Methodology:** Phornpot Chainok, Radomyos Matjiur, Rodrigo Zacca, Ricardo J. Fernandes, J. Paulo Vilas-Boas.

**Project administration:** Phornpot Chainok.

**Resources:** Phornpot Chainok, Radomyos Matjiur.

**Software:** Phornpot Chainok, Radomyos Matjiur, Jessy Lauer, Karla de Jesus, Rodrigo Zacca.

**Supervision:** Weerawat Limroongreungrat, Ricardo J. Fernandes, J. Paulo Vilas-Boas.

**Validation:** Phornpot Chainok, Radomyos Matjiur, Jessy Lauer, Weerawat Limroongreungrat, Karla de Jesus, Rodrigo Zacca, Ricardo J. Fernandes, J. Paulo Vilas-Boas.

**Visualization:** Phornpot Chainok, Radomyos Matjiur, Jessy Lauer, Weerawat Limroongreungrat, Karla de Jesus, Rodrigo Zacca, Ricardo J. Fernandes, J. Paulo Vilas-Boas.

**Writing – original draft:** Phornpot Chainok, Radomyos Matjiur, Jessy Lauer, Weerawat Limroongreungrat, Karla de Jesus, Rodrigo Zacca, Ricardo J. Fernandes, J. Paulo Vilas-Boas.

**Writing – review & editing:** Phornpot Chainok, Radomyos Matjiur, Jessy Lauer, Weerawat Limroongreungrat, Karla de Jesus, Rodrigo Zacca, Ricardo J. Fernandes, J. Paulo Vilas-Boas.

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
