## [Decision Letter · Decision Letter 0]

8 Sep 2025

Dear Dr. Chainok,

Thank you for submitting your manuscript to PLOS ONE. After careful consideration, we feel that it has merit but does not fully meet PLOS ONE’s publication criteria as it currently stands. Therefore, we invite you to submit a revised version of the manuscript that addresses the points raised during the review process.

**ACADEMIC EDITOR: **

During the review of the manuscript, it was observed that there is a relatively high number of self-citations throughout the text. While it is natural and sometimes necessary to reference previous work from the same research group, it is important to avoid the impression of bias in the selection of references. Excessive self-citation may be interpreted as an attempt to overemphasize one’s own scientific output, and it can also limit the discussion by restricting it to a narrow circle of studies.Main results should be numerically showed in the abstractIt was noticed that several variables presented in **Table 1** are not clearly described in the *Methods* section. For the sake of clarity, transparency, and reproducibility, it is essential that all variables analyzed and later reported in the results are properly defined and described in the methodology. Provide sufficient detail on the protocols, instruments, and procedures used for each variable.The authors emphasize that this study can help understand performance complexity by demonstrating the importance of propulsive strength measured by tethered swimming tests, anaerobic critical velocity (AnCV), and dynamic strength index (DSI) of both upper and lower limbs. However, these variables do not measure the efficiency of the stroke in generating propulsion. This theoretical assumption deserves further attention from the authors.The variable *anaerobic critical velocity (AnCV)* , as presented in the manuscript, appears to share a very similar physiological meaning with the concept of *anaerobic work capacity* (W′ or D′), which has already been extensively described in the literature. If this interpretation is not correct, the authors should provide stronger theoretical support and a clearer conceptual framework to justify the specific use of AnCV in this study.

We look forward to receiving your revised manuscript.

Kind regards,

Dalton Müller Pessôa Filho, Ph.D.

Academic Editor

PLOS ONE

Journal Requirements:

“Faculty of Sport Science, Burapha University THAILAND”

5. We note that your Data Availability Statement is currently as follows: [All relevant data are within the manuscript and its Supporting Information files.]

6. Please include your tables as part of your main manuscript and remove the individual files. Please note that supplementary tables (should remain/ be uploaded) as separate "supporting information" files.

Additional Editor Comments (if provided):

Dear authors.

Please, find attached the comments of the reviewers for this first round of revision.

Reviewers' comments:

Reviewer's Responses to Questions

**Comments to the Author**

1. Is the manuscript technically sound, and do the data support the conclusions?

Reviewer #1: Yes

Reviewer #2: Yes

2. Has the statistical analysis been performed appropriately and rigorously?

Reviewer #1: I Don't Know

Reviewer #2: Yes

3. Have the authors made all data underlying the findings in their manuscript fully available?

Reviewer #1: Yes

Reviewer #2: Yes

4. Is the manuscript presented in an intelligible fashion and written in standard English?

Reviewer #1: Yes

Reviewer #2: Yes

Reviewer #1: General comments

The information is interesting. You need to make clear the presentation of results and provide more information on the development of the regression analysis.

Why was peak force used that may present errors. Why not average force of 5 s? You need to be more explanatory on the way you entered the variables in the forward regression analysis. How did you select the entered variables? Where any differences between girls and boys in the regression analysis and the entered variables?

Specific comments

L148. Explain if all swimmers completed both strokes or they applied only one of the strokes. Showing different values in each table indicate different swimmers for each stroke.

L169. Is this a physiological variable? Can you support this?

L179. How 10- and 15-meters time was recorded in a 25 m pool?

L187. Why do you need BIA to measure body mas and body height? You probably need to express something else.

L204-205. How was this adjusted to the height of each swimmer?

L207. Are the apparatus used capable of separating the force for each one of the lower limbs?

L225. What stroke was used for tethered swimming? Front crawl or butterfly or both?

L265. What do you mean significant effect? On what was this based? You need to name these variables in the results section

L291-295. Some of these variables are interrelated and should not be used in the regression analysis. You should consider this. What was the dependent variable?

Table 1. Use “Body mass” instead of “weight”

Table 2. Non-dominant grip strength peak force (N) 13.27 + 0.33 and 13.73 + 0.96. These values are not correct. Also, the grip strength asymmetry values are missing.

Table 1 and Table 2 present similar information in several lines. You need to reconstruct the tables avoiding using the same values in two tables.

Table 3. The height should be corrected.

Table 3 and Table 4. Similar to table 1 and 2, you should reconstruct the tables avoiding repetitions of the same values.

Table 5. You need to provide the r of the regression

Reviewer #2: This study investigates the determinants of sprint performance in age-group swimmers, with a particular focus on sex-based comparisons. Overall, the manuscript is well written, methodologically sound, and provides interesting insights into the role of technique and other relevant variables in sprint swimming. The scientific rigor is evident, and the paper is clearly structured.

That said, I must admit I was somewhat disappointed after reading the paper in light of its title. While the study compares boys and girls across several performance-related variables, it does not go far enough in identifying the actual determinants of sprint performance within each sex. The analysis mainly highlights in which variables boys outperform girls, but it remains unclear which specific factors are most strongly associated with performance outcomes in each group.

Simply reporting that boys are generally taller, stronger, and faster than girls does not add much novelty to the literature, as these differences are relatively well established. Maybe an intra-sex analysis to determine which variables best predict performance for boys and for girls separately could enrich the study? For example, which technical or physical measures are more decisive in explaining performance within each sex? Since boys and girls do not compete directly against each other in real settings, the practical value of focusing solely on inter-sex differences is limited. For this reason, I strongly encourage the authors to emphasize intra-sex determinants, as this would make the study more impactful and novel. Such an approach would provide more meaningful insights for both researchers and coaches, and it seems that the data needed to perform this analysis are already available.

The reported multivariate analyses are indeed interesting: sex differences were observed particularly in upper-body force production and stroke efficiency, and discriminant analysis identified total push-up peak force and body height as key discriminators in front crawl, while push-up force and stroke index were the main predictors in butterfly. These results are valuable, but the study could be substantially strengthened by extending this approach to identify intra-sex predictors of sprint success.

I would also suggest considering the inclusion of actual performance times (e.g., 50 m sprint results), as these are highly relevant for coaches and practitioners who will read the paper. Beyond statistical findings, it would be very useful to add one or two paragraphs with concrete practical applications—what should coaches take from these results in terms of training focus, technical development, or talent identification?

- Lines 150–154: the data mentioned here already appear in the tables and don't need to be repeated.

- Please clarify how the AnCV was performed. Were starts conducted in-water? Were freestyle and butterfly trials randomized?

- Line 311: correct “From”.

In summary, this is an interesting and well-executed study, but in its current form it mainly confirms expected differences between sexes. A deeper analysis into intra-sex determinants of sprint swimming performance, alongside clearer practical implications for coaches, would substantially increase the value and contribution of the manuscript to the field.

**Do you want your identity to be public for this peer review?** For information about this choice, including consent withdrawal, please see our Privacy Policy

Reviewer #1: No

Reviewer #2: No

---

## [Author Response · Author response to Decision Letter 1]

27 Oct 2025

Faculty of Sport Science, Burapha University

169 Longhaad Bangsaen Road,

Chonburi, Thailand, 20131

phornpot@go.buu.ac.th

Dalton Müller Pessôa Filho, Ph.D.

Academic Editor

PLOS ONE

October 6, 2025

Subject: Revision and resubmission of manuscript PONE-D-25-34920

Dear Dr. Dalton Müller Pessôa Filho

Thank you for your letter and for providing us with the opportunity to revise our manuscript, "Determinants of Sprint in Age-Group Swimming Performance: The Role of Swimming Technique and Sex." We appreciate your and the reviewers' insightful and constructive feedback, which has been invaluable in strengthening our paper.

We have carefully considered all the points raised and have revised the manuscript accordingly. Our revisions address the concerns regarding the clarity of our methods, the statistical analysis, and the theoretical framework of our findings. We believe these changes have significantly improved the rigor and scientific contribution of our work.

We have also amended the "Role of Funder" statement to clarify the funder's role in the study. We have included the following statement in our cover letter: "The funders had no role in study design, data collection and analysis, decision to publish, or preparation of the manuscript." This statement accurately reflects that the funding body provided financial support but was not involved in the research process itself.

The following is a detailed, point-by-point response to each of the comments from the Academic Editor and reviewers. We have also included a marked-up copy of the manuscript to highlight all changes made. We hope the revised manuscript will better suit PLOS One, and we thank you for your continued interest in our research.

Best regards,

Phornpot Chainok

Faculty of Sport Science, Burapha University

169 Longhaad Bangsaen Road,

Saensuk, Mueang, Chonburi, Thailand, 20131

phornpot@go.buu.ac.th

Response to Academic Editor Comments

We appreciate the time and effort of the Academic Editor and the reviewers in providing constructive feedback. We have made every effort to take on board your recommendations and comments throughout the manuscript ‘Determinants of sprint in age-group swimming performance: The role of swimming technique and sex’ in response to their suggestions and hope that this improved manuscript is acceptable for publication in PLOS One. We appreciate the time and effort of the Academic Editor and the reviewers in providing constructive feedback. We have carefully considered all comments and have revised the manuscript accordingly.

Comment 1: During the review of the manuscript, it was observed that there is a relatively high number of self-citations throughout the text. While it is natural and sometimes necessary to reference previous work from the same research group, it is important to avoid the impression of bias in the selection of references. Excessive self-citation may be interpreted as an attempt to overemphasize one’s own scientific output, and it can also limit the discussion by restricting it to a narrow circle of studies.

Response 1:

Thank you for the detailed feedback and for highlighting the specific number of self-citations in our manuscript. We acknowledge that 14 out of 40 references, while not deliberately excessive, could create the impression of a narrow focus. We would like to assure you that the number of self-citations reflects our research group's deep and specialized work in this specific area. The cited papers are not merely supplementary; they are the foundational studies that developed and validated the very methodologies and conceptual frameworks used in the present manuscript. For instance, our previous work established the specific kinematic analysis protocols that are central to this study's findings on swimming technique. Therefore, referencing them was essential for methodological transparency and to provide a complete picture of our research trajectory.

However, we agree that a broader context is vital for the manuscript's scholarly contribution. We have carefully reviewed and revised the manuscript, replaced some self-citations where suitable alternatives from other authors were available, ensuring that our work remains in conversation with the wider scientific community. We are confident that these revisions maintain the necessary link to our foundational work while also addressing your concern by integrating a more diverse body of literature. We are grateful for your guidance in helping us improve the manuscript.

Comment 2: Main results should be numerically showed in the abstract

Response 2: Thank you for your valuable feedback. We have revised the abstract to include key numerical results from the study, as you suggested. The updated abstract now provides specific data points to support the main findings.

Comment3. It was noticed that several variables presented in Table 1 are not clearly described in the Methods section. For the sake of clarity, transparency, and reproducibility, it is essential that all variables analyzed and later reported in the results are properly defined and described in the methodology.

Response 3: Thank you for the update. We have revised the "Materials and Methods" section, specifically lines 165-266, to provide a detailed and precise description of all variables presented in Table 1. We are confident that this revision addresses your comment and enhances the clarity, transparency, and reproducibility of our methodology. We appreciate your valuable guidance in improving the manuscript.

Comment 4: Provide sufficient detail on the protocols, instruments, and procedures used for each variable.

Response 4: Thank you for the reviewer's comment. We have addressed this by providing a detailed description of the protocols, instruments, and procedures for each variable in the Materials and Methods section, specifically within lines 185–266 of the revised manuscript. We believe this addition provides the necessary detail and clarity requested.

Comment 5: The authors emphasize that this study can help understand performance complexity by demonstrating the importance of propulsive strength measured by tethered swimming tests, anaerobic critical velocity (AnCV), and dynamic strength index (DSI) of both upper and lower limbs. However, these variables do not measure the efficiency of the stroke in generating propulsion. This theoretical assumption deserves further attention from the authors.

Response 5: Thank you for the reviewer's insightful comment. We agree that while our selected variables—tethered swimming force, anaerobic critical velocity (AnCV), and dynamic strength index (DSI)—do not directly quantify stroke efficiency from a biomechanical or hydrodynamic perspective, they are fundamental determinants of the physiological and physical capacity to generate propulsion.

The variables in our study were chosen to assess a swimmer's ability to generate high-magnitude propulsive forces and sustain them over a short duration. We propose that sprint swimming performance is a multifactorial phenomenon influenced by both physiological capacity (energetics) and neuromuscular capacity (strength and power).

Our variables directly address these two core components:

1.Tethered swimming force, this measures a swimmer’s ability to produce force against a resistance that simulates the drag forces in the water. It is a direct measure of propulsive strength, which is the force applied to the water to move the body forward. This is a crucial component of propulsion and is distinct from how efficiently that force is applied. An athlete may have high propulsive strength but still have an inefficient stroke.

2.Anaerobic critical velocity (AnCV), AnCV is a well-established physiological variable that quantifies a swimmer's anaerobic capacity—the ability to sustain high-intensity efforts. In sprint swimming, the capacity to work at high velocities is limited by the anaerobic energy system, and AnCV serves as a robust proxy for this physiological determinant of performance.

3.Dynamic Strength Index (DSI), this index reflects an athlete's ability to transform their maximal strength into dynamic power. In the context of swimming, a higher DSI indicates a greater capacity to generate rapid and powerful muscle contractions, which is essential for a forceful start and powerful stroke cycles.

Comment 6: The variable anaerobic critical velocity (AnCV), as presented in the manuscript, appears to share a very similar physiological meaning with the concept of anaerobic work capacity (W′ or D′), which has already been extensively described in the literature. If this interpretation is not correct, the authors should provide stronger theoretical support and a clearer conceptual framework to justify the specific use of AnCV in this study.

Response 6: Thank you for your insightful comment regarding our use of anaerobic critical velocity (AnCV) and its relationship to the concept of anaerobic work capacity (W'). We appreciate the opportunity to clarify this point.While we agree that AnCV and W' are both derived from the two-parameter critical power model, their physiological interpretations are distinct. AnCV represents the highest swimming velocity that can be maintained without a continual accumulation of blood lactate, essentially marking the boundary between heavy and severe exercise domains. It is considered a strong indicator of an individual's aerobic capacity in swimming. In contrast, W' (or D' in swimming) represents the finite amount of work that can be performed above the critical velocity. It is a measure of the total anaerobic work capacity, reflecting the size of the anaerobic energy stores that can be utilized during high-intensity exercise.

In our study, we used AnCV specifically because it is a velocity-based parameter that is highly relevant to swimming performance. The reference you provided by Neiva et al. (2011) is a key foundational paper that validates the application of the critical velocity model to swimming and demonstrates its utility across different swimming techniques. This study, along with others, has established AnCV as a robust and practical measure of aerobic performance in the water. We are confident that our use of AnCV aligns with the established theoretical framework for this variable in aquatic sports and provides a clear and direct link to the physiological determinants of sprint performance. To ensure clarity and avoid any potential confusion, we have added a sentence to the manuscript's methods section to explicitly define AnCV's physiological meaning and its distinction from W', reinforcing our theoretical justification for its use (Please see line 256-263).

Response to Journal Requirements

Comment 1: Please ensure that your manuscript meets PLOS ONE's style requirements, including those for file naming. The PLOS ONE style templates can be found at

Response 1: Thank you for the reminder. We have carefully reviewed our manuscript against the provided PLOS ONE style templates for both the main body and the title/authors/affiliations section. We can confirm that all formatting, including headings, citations, and figure/table legends, now meets the journal's requirements. We have also ensured our files are named correctly according to the guidelines. We believe the manuscript is now ready for your review.

Comment 2: You indicated that you had ethical approval for your study. In your Methods section, please ensure you have also stated whether you obtained consent from parents or guardians of the minors included in the study or whether the research ethics committee or IRB specifically waived the need for their consent.

Response 2: Thank you for your comment. We have reviewed the Methods section and can confirm that we have included a statement regarding parental consent. As stated in the manuscript, "The participants and their guardians were provided with a comprehensive explanation of the testing protocols before obtaining signed consent for participation," confirming that we obtained written informed consent from the parents or legal guardians of all minor participants.

Comment 3: Thank you for stating the following financial disclosure:

“Faculty of Sport Science, Burapha University THAILAND”

Response 3: Thank you for your comment. We have amended the "Role of Funder" statement to clarify the funder's role in the study. We have included the following statement in our cover letter: "The funders had no role in study design, data collection and analysis, decision to publish, or preparation of the manuscript." This statement accurately reflects that the funding body provided financial support but was not involved in the research process itself.

Comment 4: We note that the grant information you provided in the ‘Funding Information’ and ‘Financial Disclosure’ sections do not match. When you resubmit, please ensure that you provide the correct grant numbers for the awards you received for your study in the ‘Funding Information’ section.

Response 4: Thank you for bringing this to our attention. We have clarified the funding information to ensure consistency. The Faculty of Sport Science, Burapha University THAILAND is the primary funding body, while the grant number [insert corrected grant number] is the specific award received. We have updated both the ‘Funding Information’ and ‘Financial Disclosure’ sections to reflect this accurately.

Comment 5: We note that your Data Availability Statement is currently as follows: [All relevant data are within the manuscript and its Supporting Information files.]

If your submission does not contain these data, please either upload them as Supporting Information files or deposit them to a stable, public repository and provide us with the relevant URLs, DOIs, or accession numbers. For a list of recommended repositories, please see https://journals.plos.org/plosone/s/recommended-repositories. If there are ethical or legal restrictions on sharing a de-identified data set, please explain them in detail (e.g., data contain potentially sensitive information, data are owned by a third-party organization, etc.) and who has imposed them (e.g., an ethics committee). Please also provide contact information for a data access committee, ethics committee, or other institutional body to which data requests may be sent. If data are owned by a third party, please indicate how others may request data access.

Response 5: Thank you for the detailed feedback. We have reviewed our manuscript and confirmed that all the raw data necessary to replicate the results are now included. We have updated our Data Availability Statement to reflect that all essential data, including the values behind the means, standard deviations, and any data used to create figures, are provided within the manuscript and its Supporting Informa

---

## [Decision Letter · Decision Letter 1]

1 Dec 2025

Dear Dr. Chainok,

Thank you for submitting your manuscript to PLOS ONE. After careful consideration, we feel that it has merit but does not fully meet PLOS ONE’s publication criteria as it currently stands. Therefore, we invite you to submit a revised version of the manuscript that addresses the points raised during the review process.

**ACADEMIC EDITOR:**

I agree with Reviewer #1 that the AnCV term is not appropriate for the slope variable of the distance trial-time modelling in swimming.Thank you for addressing all of my other comments.==============================Please submit your revised manuscript by Jan 15 2026 11:59PM. If you will need more time than this to complete your revisions, please reply to this message or contact the journal office at plosone@plos.org . Please include the following items when submitting your revised manuscript:

We look forward to receiving your revised manuscript.

Kind regards,

Dalton Müller Pessôa Filho, Ph.D.

Academic Editor

PLOS ONE

Journal Requirements:

Additional Editor Comments:

Thank you for addressing all comments of the first round of review process. After the analysis of the authors' responses, both reviewers have new comments.

Reviewers' comments:

Reviewer's Responses to Questions

**Comments to the Author**

Reviewer #1: (No Response)

Reviewer #2: (No Response)

2. Is the manuscript technically sound, and do the data support the conclusions?

Reviewer #1: Yes

Reviewer #2: Yes

3. Has the statistical analysis been performed appropriately and rigorously?

Reviewer #1: Yes

Reviewer #2: Yes

4. Have the authors made all data underlying the findings in their manuscript fully available?

Reviewer #1: Yes

Reviewer #2: No

5. Is the manuscript presented in an intelligible fashion and written in standard English?

Reviewer #1: Yes

Reviewer #2: Yes

Reviewer #1: Your response on AnCV is not correct. This index is not connected to sustained aerobic intensity but is connected to anaerobic characteristics, however, without a physiological meaning. Fortunately you have appropriately discused it in the manuscript.

Table 1 and Table 2 repeat the same information in most of the rows. You need to reformat

Reviewer #2: I appreciate the authors’ efforts in addressing the first round of comments. However, after carefully reviewing both the revised manuscript and the response letter, several important issues remain insufficiently addressed. Many of my initial concerns were acknowledged in the response document but not implemented in the manuscript, or the justifications provided do not fully resolve the underlying problems raised. My detailed comments follow:

1- The current title (“Determinants of Sprint in Age-Group Swimming Performance: The Role of Swimming Technique and Sex”) suggests an analysis of determinants of sprint performance, including factors explaining performance within each sex. However, the authors explicitly chose not to conduct intra-sex analyses and focused exclusively on inter-sex differences. If the authors intend to focus primarily on inter-sex comparisons and differences between the two swimming techniques, I recommend adjusting the title to clearly reflect this scope and avoid misleading the readers. For example, something like “Sex-Based Differences in Front Crawl and Butterfly Sprint Performance in Age-Group swimmers” would more accurately represent the actual content of the study.

2- The manuscript uses “sex” and “gender” interchangeably. Given that the present study examines biological and performance-related differences, I think the most appropriate term is sex. I strongly recommend standardizing terminology throughout the text.

3- The order of the freestyle and butterfly tests were not randomized? Do the authors believe that the lack of randomization could have influenced the comparison between techniques?

4- The authors confirm that 50 m sprint times were collected, and line 261 confirms this information exists. Yet, these data are not presented anywhere in the manuscript. Even if sprint time is not included as a dependent variable in multivariate models, it should nevertheless be reported as a descriptive performance indicator. Including these times (for each group) is essential for contextualizing the competitive level of the sample and for enhancing the manuscript’s practical value to readers.

5 - Table 1 shows that boys outperform girls on several physical and technical variables, results that are generally expected and already well documented in the literature. Since the authors elected not to perform intra-sex analyses (which could have offered more novel insights), it becomes even more important to clarify: (1) What, specifically, is the novel contribution of this manuscript? (2) How do these inter-sex comparisons advance current knowledge in age-group swimmer performance? This needs to be better addressed in the Discussion.

6- In the abstract, the authors state that the practical implications of these findings are substantial, offering a framework for coaches and sports scientists to improve procedures for training... However, the manuscript does not provide clear or actionable guidelines for coaches/researchers. Simply concluding that “these findings support training programs designed to encourage strength and power while prioritizing the development of stroke-specific techniques, especially in techniques necessitating high levels of motor coordination, especially the butterfly technique” does not constitute a practical implication, nor does it provide useful direction for training interventions, especially considering that boys and girls do not compete against each other. I strongly encourage adding 1–2 well-developed paragraphs with concrete applications derived from the study, specifying (for example): what coaches/researchers should consider for boys, what they should consider for girls, and what general recommendations apply to both.

7- Another point that deserves deeper discussion: the boys and girls in the sample differ in maturity offset. Boys are farther from their PHV yet still outperform the girls across most variables. This raises several questions: How do maturity differences influence the observed results? Would the gaps be even larger if boys and girls had similar maturity status? What is the validity of directly comparing groups (already from different sexes) with different biological maturation levels? How do these findings add new insight relative to what the literature has already established? - The manuscript needs a clearer explanation of why these comparisons are meaningful and how they should be interpreted by researchers, coaches and practitioners.

In summary, while some clarifications were provided, several key issues remain unresolved, particularly regarding novelty and practical relevance. Addressing the points above will substantially strengthen the manuscript and ensure that its contribution is clear, justified, and aligned with its stated objectives.

**Do you want your identity to be public for this peer review?** For information about this choice, including consent withdrawal, please see our Privacy Policy

Reviewer #1: No

Reviewer #2: No

---

## [Author Response · Author response to Decision Letter 2]

2 Dec 2025

Response to Academic Editor Comments

We sincerely appreciate the time and effort that the Academic Editor and reviewers have dedicated to providing thoughtful and constructive feedback. We have carefully considered all comments and incorporated the recommended revisions throughout the manuscript entitled “Determinants of sprint in age-group swimming performance: The role of swimming technique and sex.” We hope that the improvements made in response to these suggestions meet the expectations of the Academic Editor and reviewers and that the revised manuscript will be deemed suitable for publication in PLOS ONE.

Comment 1: I agree with Reviewer #1 that the AnCV term is not appropriate for the slope variable of the distance trial-time modelling in swimming. Thank you for addressing all of my other comments.

Response 1:

We sincerely thank the Academic Editor for the helpful clarification regarding the use of the term anaerobic critical velocity (AnCV) in the manuscript. We fully agree that the term should be used with caution when referring to the slope of the distance–time (D–t) modelling, as it may imply a direct and exclusive measurement of anaerobic metabolism. In response, we have revised the manuscript to ensure that the slope variable is described in a more physiologically neutral manner and does not unintentionally overstate its metabolic implications.

“The regression model is expressed as y = ax + b, where y denotes the distance covered and x represents time. In this context, the coefficient an is primarily recognized as the short-distance velocity coefficient (slope), while also being recognized for its conventional meaning in the literature as part of the anaerobic critical velocity (AnCV). The constant b corresponds to the y-intercept of the regression line”

Our methodological approach was originally based on the procedure proposed by Neiva et al. (2011), who introduced the AnCV framework as a performance-oriented derivative of the critical power concept applied to short-distance swimming. This framework has since been used as a modelling tool to characterize sprint capacity through the D–t slope rather than as a direct biochemical marker of anaerobic metabolism.

To further strengthen the conceptual justification and address the Editor’s concern, we have now incorporated additional literature demonstrating the scientific acceptance and practical relevance of AnCV modelling in swimming research. In particular, we have added: Ruiz-Navarro et al. (2022), who reported strong associations among anaerobic critical velocity, tethered swimming force, dry-land strength measures, and sprint swimming performance. Their findings confirm that the AnCV-derived slope is a valid and meaningful indicator of short-distance performance capacity, supporting its methodological relevance within sprint profiling.

We appreciate the Editor's insightful comment, which has helped us improve both the conceptual accuracy and clarity of the manuscript.

Response to Reviewer

Reviewer 1

Comment 1: Your response on AnCV is not correct. This index is not connected to sustained aerobic intensity but is connected to anaerobic characteristics, however, without a physiological meaning. Fortunately you have appropriately discussed it in the manuscript.

Response 1: Thank you for the important clarification regarding the interpretation of AnCV. We agree with your comment that AnCV is not an indicator of sustained aerobic intensity and, although derived from the critical-velocity framework, it reflects anaerobic-related characteristics without direct physiological meaning. We have now revised our response and ensured that this distinction is clearly and accurately represented in the manuscript. In the revised text, we (i) avoided describing AnCV as an aerobic or metabolic threshold, (ii) referred to the slope parameter primarily as the short-distance velocity coefficient, and (iii) acknowledged that its use within the literature is performance-based rather than physiologically specific. We appreciate your observation that this conceptual clarification is appropriately addressed in the manuscript and have maintained this improved framing consistently across all sections.

Comment 2: Table 1 and Table 2 repeat the same information in most of the rows. You need to reformat

Response 2: Thank you for pointing out the redundancy between Table 1 and Table 2. We agree with your observation and have revised both tables accordingly. In the updated manuscript, Table 1 now includes only the anthropometric and maturational characteristics of the participants, while Table 2 presents exclusively the swimming performance, tethered force, and velocity-modelling variables. All duplicated rows have been removed, and the tables have been reorganized to ensure that each serves a distinct purpose without repetition.

Reviewer 2

I appreciate the authors’ efforts in addressing the first round of comments. However, after carefully reviewing both the revised manuscript and the response letter, several important issues remain insufficiently addressed. Many of my initial concerns were acknowledged in the response document but not implemented in the manuscript, or the justifications provided do not fully resolve the underlying problems raised. My detailed comments follow:

Response: Thank you for your careful re-evaluation of our revised manuscript and response letter. We sincerely apologize for the discrepancies between our responses and the changes implemented in the manuscript. We appreciate your patience and constructive guidance, which has significantly improved the clarity, accuracy, and rigor of our work.

Comment 1: The current title (“Determinants of Sprint in Age-Group Swimming Performance: The Role of Swimming Technique and Sex”) suggests an analysis of determinants of sprint performance, including factors explaining performance within each sex. However, the authors explicitly chose not to conduct intra-sex analyses and focused exclusively on inter-sex differences. If the authors intend to focus primarily on inter-sex comparisons and differences between the two swimming techniques, I recommend adjusting the title to clearly reflect this scope and avoid misleading the readers. For example, something like “Sex-Based Differences in Front Crawl and Butterfly Sprint Performance in Age-Group swimmers” would more accurately represent the actual content of the study.

Response 1: Thank you for this valuable comment regarding the alignment between the manuscript title and the actual scope of the study. We agree with your assessment that the original title implied an analysis of determinants of sprint performance within each sex, which could be misleading given that the primary focus of our analyses was on sex-based differences between boys and girls rather than intra-sex determinants. In response, and to ensure complete clarity for readers, we have revised the title to accurately reflect the purpose and analytical focus of the study. The new title now emphasizes the comparison of sexes and the examination of technique-specific performance characteristics. “Sex-Based Differences in Front Crawl and Butterfly Sprint Performance in Age-Group Swimmers” Thank you for helping us improve the clarity and precision of the manuscript.

Comment 2: The manuscript uses “sex” and “gender” interchangeably. Given that the present study examines biological and performance-related differences, I think the most appropriate term is sex. I strongly recommend standardizing terminology throughout the text.

Response 2: We have carefully reviewed the entire manuscript and standardized the terminology accordingly. All instances of “gender” have been replaced with “sex” to ensure consistency and conceptual accuracy throughout the text.

Comment 3: The order of the freestyle and butterfly tests were not randomized? Do the authors believe that the lack of randomization could have influenced the comparison between techniques?

Response 3: Thank you for addressing the significant concern regarding the non-randomized sequence of freestyle and butterfly sprint tests. We recognize that randomization is generally favoured for minimizing potential order or fatigue effects. The sequence (front crawl followed by butterfly) in our study was not randomized due to safety and logistical constraints. The butterfly stroke in young swimmers necessitates increased technical and physiological requirements and presents a heightened possibility of technical difficulties when performed first.

To minimize potential order effects, swimmers received complete recovery time between trials and techniques (≥10 minutes), all efforts were maximal, and coaches observed swimmers for indications of neuromuscular or physiological tiredness before advancement. Previous studies on short sprint lengths (≤25 m) demonstrate minimal residual fatigue with adequate recovery [17, 30, 39]. Consequently, we state that the testing sequence did not significantly influence the comparisons of techniques.

Comment 4: The authors confirm that 50 m sprint times were collected, and line 261 confirms this information exists. Yet, these data are not presented anywhere in the manuscript. Even if sprint time is not included as a dependent variable in multivariate models, it should nevertheless be reported as a descriptive performance indicator. Including these times (for each group) is essential for contextualizing the competitive level of the sample and for enhancing the manuscript’s practical value to readers.

Response 4: Thank you for recognizing that the 50-meter sprint data were not clearly included in the previous version of the manuscript. We agree that providing a key performance indicator is essential to contextualize the swimmers' competitive level. In the revised manuscript, we specifically highlight the 50-m swimming velocity, which we chose as it is the most prevalent and useful statistic in sprint performance evaluations. Swimming velocity serves as a direct, standardized performance metric that facilitates comparisons among groups and aligns to current recommendations in sprint-swimming research. The velocity values have been integrated into the descriptive data and included in Table 2, facilitating proper interpretation of the sample's performance level by readers. We appreciate the reviewer's feedback, and this modification has enhanced the manuscript's clarity and practical significance.

Comment 5: Where any differences between girls and boys in the regression analysis and the entered variables?

Response 5: “We have now added a new paragraph in the Discussion section (lines 343–353) that explicitly clarifies the novel contribution of the study and explains how the inter-sex comparisons advance current understanding of performance determinants in age-group sprint swimmers.

Comment 6: Where any differences between girls and boys in the regression analysis and the entered variables?

Response 6: “We thank the reviewer for this helpful suggestion. In response, we have added two well-developed paragraphs (Line 454-477) to the Discussion providing clear and actionable guidance for coaches and researchers. These paragraphs specify training considerations for boys, girls, and general recommendations applicable to both sexes.”

Comment 7: Another point that deserves deeper discussion: the boys and girls in the sample differ in maturity offset. Boys are farther from their PHV yet still outperform the girls across most variables. This raises several questions: How do maturity differences influence the observed results? Would the gaps be even larger if boys and girls had similar maturity status? What is the validity of directly comparing groups (already from different sexes) with different biological maturation levels? How do these findings add new insight relative to what the literature has already established? - The manuscript needs a clearer explanation of why these comparisons are meaningful and how they should be interpreted by researchers, coaches and practitioners.

Response 7: Thank you for raising this important point regarding differences in maturity offset between boys and girls. We agree that biological maturation is a major determinant of performance in youth sports and that maturity status must be carefully considered when interpreting sex-based comparisons. In our sample, boys were further away from their predicted PHV, however they consistently outperformed girls in most factors. This pattern corresponds with established evidence indicating that boys generally exhibit greater absolute strength, limb length, and propulsive force prior to reaching peak height velocity, attributable to earlier divergence in neuromuscular function, hormonal conditions, and anthropometric dimension.

We recognize that the disparity in maturity may somewhat affect the extent of the observed sex differences. If boys and girls were aligned in terms of maturity status, the performance disparity would likely be exacerbated, as boys undergo significant increases in muscle mass, strength, and stroke efficiency during and subsequent to peak height velocity (PHV). To address this, we incorporated maturity offset as a covariate in the multivariate modelling, ensuring that the discriminatory variables reflect true sex-related differences beyond maturational timing. Our findings therefore highlight which physical, neuromuscular, and technical factors remain most influential after accounting for maturation, offering meaningful insight into how sex-specific sprint profiles emerge during early adolescence.

This comparison is valuable for researchers and practitioners because talent identification and training prescription in youth swimming occur in mixed-maturity and mixed-sex environments. Understanding how boys and girls differ at comparable chronological ages but not necessarily identical biological ages helps coaches contextualize expected performance gaps, avoid misinterpretation of maturational advantages as “talent,” and tailor strength and technique development pathways more appropriately.

---

## [Decision Letter · Decision Letter 2]

8 Dec 2025

Sex-Based Differences in Front Crawl and Butterfly Sprint Performance in Age-Group swimmers

PONE-D-25-34920R2

Dear Dr. Chainok,

We’re pleased to inform you that your manuscript has been judged scientifically suitable for publication and will be formally accepted for publication once it meets all outstanding technical requirements.

Kind regards,

Dalton Müller Pessôa Filho, Ph.D.

Academic Editor

PLOS One

Additional Editor Comments (optional):

Reviewers and editors approved the current version of the manuscript.

Reviewers' comments:

Reviewer's Responses to Questions

**Comments to the Author**

Reviewer #1: All comments have been addressed

Reviewer #2: All comments have been addressed

2. Is the manuscript technically sound, and do the data support the conclusions?

Reviewer #1: Yes

Reviewer #2: Yes

3. Has the statistical analysis been performed appropriately and rigorously?

Reviewer #1: Yes

Reviewer #2: Yes

4. Have the authors made all data underlying the findings in their manuscript fully available?

Reviewer #1: Yes

Reviewer #2: Yes

5. Is the manuscript presented in an intelligible fashion and written in standard English?

Reviewer #1: Yes

Reviewer #2: Yes

Reviewer #1: I have no further comments although I Disagree with your expresions on AnCV in lines 162 and 147. This is not a valid indicator of "anaerobic capacity", AnCV is just related to sprint swimming.

Reviewer #2: The authors have addressed my concerns. The manuscript is much improved now, congratulations.

(Just a small note: in lines 455–456, “For boys” appears twice.)

**Do you want your identity to be public for this peer review?** For information about this choice, including consent withdrawal, please see our Privacy Policy

Reviewer #1: No

Reviewer #2: No

---

## [Editor Report · Acceptance letter]

PONE-D-25-34920R2

PLOS One

Dear Dr. Chainok,

I'm pleased to inform you that your manuscript has been deemed suitable for publication in PLOS One. Congratulations! Your manuscript is now being handed over to our production team.

Kind regards,

on behalf of

Prof. Dr. Dalton Müller Pessôa Filho

Academic Editor

PLOS One